# Mapping Disorders with Neurological Features Through Mitochondrial Impairment Pathways: Insights from Genetic Evidence

**DOI:** 10.3390/cimb47070504

**Published:** 2025-07-01

**Authors:** Anna Makridou, Evangelie Sintou, Sofia Chatzianagnosti, Iasonas Dermitzakis, Sofia Gargani, Maria Eleni Manthou, Paschalis Theotokis

**Affiliations:** 1Department of Histology-Embryology, School of Medicine, Aristotle University of Thessaloniki, 54124 Thessaloniki, Greece; annmak4@gmail.com (A.M.); evangelisintou@gmail.com (E.S.); sofiachatzianagnosti@gmail.com (S.C.); sgargani@bio.auth.gr (S.G.); mmanthou@auth.gr (M.E.M.); ptheotokis@auth.gr (P.T.); 22nd Department of Neurology University General Hospital AHEPA, Medical School, Aristotle University of Thessaloniki, 54636 Thessaloniki, Greece

**Keywords:** mitochondrial diseases, neurological manifestations, mitochondrial genetics, metabolic pathway disruption, mitochondrial dysfunction in neurodegeneration, precision medicine, mitochondrial replacement therapy

## Abstract

Mitochondrial dysfunction is a key driver of neurological disorders due to the brain’s high energy demands and reliance on mitochondrial homeostasis. Despite advances in genetic characterization, the heterogeneity of mitochondrial diseases complicates diagnosis and treatment. Mitochondrial dysfunction spans a broad clinical spectrum, from early-onset encephalopathies to adult neurodegeneration, with phenotypic and genetic variability necessitating integrated models of mitochondrial neuropathology. Mutations in nuclear or mitochondrial DNA disrupt energy production, induce oxidative stress, impair mitophagy and biogenesis, and lead to neuronal degeneration and apoptosis. This narrative review provides a structured synthesis of current knowledge by classifying mitochondrial-related neurological disorders according to disrupted biochemical pathways, in order to clarify links between genetic mutations, metabolic impairments, and clinical phenotypes. More specifically, a pathway-oriented framework was adopted that organizes disorders based on the primary mitochondrial processes affected: oxidative phosphorylation (OXPHOS), pyruvate metabolism, fatty acid β-oxidation, amino acid metabolism, phospholipid remodeling, multi-system interactions, and neurodegeneration with brain iron accumulation. Genetic, clinical and molecular data were analyzed to elucidate shared and distinct pathophysiological features. A comprehensive table synthesizes genetic causes, inheritance patterns, and neurological manifestations across disorders. This approach offers a conceptual framework that connects molecular findings to clinical practice, supporting more precise diagnostic strategies and the development of targeted therapies. Advances in whole-exome sequencing, pharmacogenomic profiling, mitochondrial gene editing, metabolic reprogramming, and replacement therapy—promise individualized therapeutic approaches, although hurdles including heteroplasmy, tissue specificity, and delivery challenges must be overcome. Ongoing molecular research is essential for translating these advances into improved patient care and quality of life.

## 1. Introduction

The central nervous system (CNS), despite accounting for only approximately 2% of total body mass, consumes over 20% of the body’s energy resources [1]. This disproportionate energy demand renders it highly dependent on mitochondrial function for maintaining physiological activity and homeostasis. Neurons require continuous and finely regulated energy production to support key processes such as synaptic transmission, ion homeostasis, and axonal transport. Mitochondria are critical to CNS integrity as dynamic and multifunctional organelles central to cellular metabolism. Beyond ATP generation, they regulate calcium buffering, redox signaling, apoptotic pathways, and the biosynthesis of metabolic intermediates. This elevated reliance on mitochondrial function makes the nervous system particularly susceptible to mitochondrial dysfunction, often resulting in severe and irreversible neurological impairment. At the physiological level, mitochondria sustain essential neuronal activities, including synaptic plasticity and neurotransmission, by supplying energy for cytoskeletal remodeling, vesicular trafficking, and signal transduction. Calcium (Ca^2+^) homeostasis, modulated in part by mitochondrial buffering, is crucial for the activation of Ca^2+^-dependent enzymes and transcription factors involved in neuronal development and adaptation. Mitochondrial biogenesis, tightly regulated through transcriptional and translational coordination between nuclear and mitochondrial genomes, ensures sufficient ATP production to meet localized energy demands within neurons [2]. In parallel, mitochondrial dynamics—fission and fusion processes—are vital for organelle distribution, quality control, and adaptation to cellular stressors.

Disruptions in mitochondrial physiology can stem from mutations in either nuclear DNA (nDNA) or mitochondrial DNA (mtDNA), impairing essential bioenergetic, metabolic, or structural functions [3]. A critical aspect of mitochondrial-related neurological disorders is the complex and often unpredictable relationship between genotype and phenotype, shaped by both the origin of the mutation—mitochondrial or nuclear—and its mode of inheritance. Mutations in mitochondrial DNA (mtDNA) typically follow maternal inheritance and can result in heteroplasmy, where mutant and wild type mtDNA coexist within cells. Heteroplasmy, the coexistence of normal and mutated mitochondrial DNA within cells, contributes to the unpredictable severity and distribution of symptoms across individuals and tissues. The proportion of mutant mtDNA can vary across tissues and over time, influencing disease severity, age of onset, and symptom heterogeneity [4,5]. In contrast, nuclear-encoded mitochondrial gene mutations adhere to Mendelian inheritance patterns—autosomal dominant, autosomal recessive, or X-linked—allowing clearer, though not always straightforward, predictions of disease risk and transmission. Importantly, identical mutations in either mtDNA or nDNA can give rise to divergent clinical outcomes depending on background genetic modifiers, tissue energy demands, and environmental influences. For example, biallelic POLG mutations can lead to a spectrum of phenotypes ranging from childhood-onset epilepsy to adult-onset ataxia or myopathy, demonstrating how nuclear gene defects can have pleiotropic effects [6]. Understanding these genotype–phenotype correlations and their inheritance frameworks is essential for accurate diagnosis, risk assessment, and genetic counseling, as well as for identifying patients who may benefit from emerging precision therapies. Furthermore, expanding genomic databases and functional annotation of variants continue to refine our ability to map specific genetic defects to distinct neurological presentations, paving the way for precision medicine in mitochondrial diseases.

Despite increasing recognition of mitochondria as key regulators of cellular metabolism, integrative frameworks remain lacking that link specific mitochondrial metabolic dysfunctions to disorders with neurological features. Much of the existing literature concentrates on isolated metabolic pathways or individual conditions, often overlooking the broader context of mitochondrial impairment across disease spectra. This fragmented perspective can hinder diagnostic precision and therapies’ development. To narrow the gap, the present narrative review proposes a pathway-oriented approach to analytical and tabular mapping disorders with neurological involvement, grounded in the specific mitochondrial processes affected. These include oxidative phosphorylation, pyruvate metabolism, fatty acid β-oxidation, amino acid metabolism, phospholipid remodeling, multi-system mitochondrial functions, and mechanisms related to neurodegeneration with brain iron accumulation. This review provides a structured, genetically informed map of mitochondrial pathways linked to neurological disorders, offering new avenues for diagnostics and precision therapies, ultimately supporting advances in mechanistic understanding and precision medicine.

## 2. Mitochondrial Disorders Categorized by Metabolic Pathways

Mitochondria are essential organelles responsible for energy production, metabolic regulation, and cellular signaling. Dysfunction of these organelles can lead to significant health consequences, particularly in tissues with high energy demands such as the central nervous system. The complexity of mitochondrial diseases is attributed to their genetic basis, involving both nuclear DNA and mitochondrial DNA, as well as the wide range of metabolic pathways in which mitochondria are involved. Although many of these disorders present with overlapping clinical features, their molecular mechanisms differ depending on the specific metabolic pathway affected. Classifying mitochondrial diseases according to the disrupted pathway provides a clearer understanding of their underlying biology and supports the development of targeted therapeutic approaches. The principal pathways include oxidative phosphorylation, pyruvate metabolism, fatty acid β-oxidation, amino acid metabolism, phospholipid remodeling, and iron regulation in the brain. The following section presents a detailed classification of mitochondrial disorders based on the primary metabolic pathway involved.

Disruptions in these metabolic pathways—particularly those affecting oxidative phosphorylation, pyruvate metabolism, fatty acid β-oxidation, and others—can severely impair mitochondrial function, leading to energy deficits, oxidative stress, and the activation of apoptotic pathways. For instance, calcium overload can trigger the opening of the mitochondrial permeability transition pore (mPTP), releasing pro-apoptotic factors such as cytochrome c, as well as apoptosis-inducing factor (AIF), SMAC/DIABLO, and Endonuclease G, which together mediate both caspase-dependent and independent cell death pathways [7,8,9,10]. While mitochondrial quality control mechanisms, including mitophagy and biogenesis, ordinarily maintain organellar integrity, these processes may become inadequate in the context of chronic stress or genetic defects, driving mitochondrial dysfunction and cellular pathology. Neurological features arising from mitochondrial impairment exhibit a broad clinical spectrum, from early-onset encephalopathies to adult-onset neurodegenerative diseases. Representative conditions include Leigh syndrome, MELAS, congenital lactic acidosis, Sengers syndrome, MEGDEL syndrome, and fragile X syndrome. The phenotypic and genetic heterogeneity observed across these disorders reflects the complex biology of mitochondria and their essential role in neuronal survival and plasticity. Moreover, overlapping clinical features among disorders affecting distinct metabolic pathways highlight the necessity for an integrated understanding of mitochondrial involvement in neuropathology [2].

### 2.1. OXPHOS Pathway

The OXPHOS is the cornerstone of mitochondrial energy production. It involves the electron transport chain (ETC), which generates by transferring electrons from NADH and FADH_2_ to oxygen, coupled with proton pumping across the inner mitochondrial membrane, as we can see in Figure 1. Dysfunctions in OXPHOS result in impaired ATP synthesis and increased oxidative stress [11].

One of the most notable examples of the OXPHOS pathway dysfunction is Leigh syndrome, a neurodegenerative disorder caused by mutations in over 75 genes [12]. The genes that result to Leigh syndrome are ETC—related genes and include MT-ND (Complex I), SURF1 (Complex IV), ATP6 (Complex V), NDUFS1 (Complex I), SDHA (Complex II), UQCRQ (Complex III), COX10 (Complex IV), and ATP5E (Complex V). Additionally, defects in genes such as PDHA1, PDHB, COQ2, and COQ9, can be involved. The most common gene causing Leigh syndrome is SURF1. Loss-of-function mutations in SURF1 impair complex IV activity, leading to reduced ATP production, increased lactate accumulation, and neuronal energy failure. This energy deficit is particularly detrimental in high-demand tissues like the brainstem and basal ganglia, resulting in the neurodegenerative features characteristic of Leigh syndrome—such as developmental regression, hypotonia, respiratory abnormalities, and brain lesions visible on MRI [12,13].

Similarly, Alpers-Huttenlocher Syndrome (AHS) though primarily rooted in defective mtDNA replication due to *POLG* mutations, ultimately disrupts OXPHOS by impairing the synthesis of ETC components encoded in mtDNA. More specifically, POLG gene encodes the catalytic subunit of mtDNA polymerase gamma, an enzyme essential for mtDNA replication and maintenance. The severity of AHS varies; compound heterozygous mutations are often associated with more severe, early-onset phenotypes, while certain homozygous mutations can result in later-onset or milder presentations. This is thought to occur because compound heterozygous mutations may affect different functional domains of the POLG protein, leading to a broader or more synergistic loss of function. In contrast, some homozygous mutations are hypomorphic, preserving partial enzyme activity. The precise mechanisms behind this phenotypic variability remain incompletely understood, though the mutation’s position and effect on POLG function likely play a role [14,15].

A comparable mechanism underlies Kearns-Sayre Syndrome (KSS), which arises from large-scale deletions in mtDNA affecting multiple ETC genes simultaneously. Kearns-Sayre Syndrome (KSS) is a clinical subtype of chronic progressive external ophthalmoplegia (CPEO). It is maternally inherited and is characterized by a distinct triad: onset before the age of 20, CPEO, and pigmentary retinopathy. Approximately 90% of KSS cases are sporadic, arising from large-scale deletions of mtDNA, typically ranging from 1.1 to 10 kilobases in size. The most common deletion is the 4977-base pair “common deletion”, which accounts for more than one-third of KSS cases. The molecular genetics of KSS are primarily associated with these large-scale deletions or rearrangements in mtDNA, which typically disrupt genes involved in the ETC and oxidative OXPHOS—processes crucial for mitochondrial energy production in the form of ATP. The deletions can affect several key mitochondrial genes, impairing their function and leading to a loss of mitochondrial activity. This results in energy production deficits, particularly in high-energy-demand tissues, and contributes to the progressive symptoms seen in KSS [16,17,18].

In disorders such as Mitochondrial Encephalomyopathy, Lactic Acidosis, and Stroke-like episodes (MELAS) Syndrome the pathology centers around defective mitochondrial tRNA. More specifically, the dysregulation is caused primarily due to mtDNA mutations (e.g., m.3243A > G) in the MT-TL1 gene affecting Complex I or tRNA (tRNA-Leu), leads to OXPHOS dysfunction. This tRNA is crucial for the translation of mitochondrial proteins necessary for energy production. The m.3243A > G mutation impairs mitochondrial translation, affecting the synthesis of subunits in the ETC and leading to decreased energy production. The resulting energy deficiency contributes to the multi-organ dysfunction seen in MELAS [19].

Likewise, Myoclonic Epilepsy with Ragged-Red Fibers (MERRF Syndrome) is often caused by mutations in mitochondrial tRNA, affecting overall OXPHOS. This rare mitochondrial disorder is mainly caused by mutations in the tRNA (Lys) gene (such as m.8344A > G, m.8356T > C, and m.8363G > A), with less common mutations in other tRNA genes like MT-TL1, tRNA (Ile), MT-TF, or MT-TP. These mutations impair the proper translation of mitochondrial genes crucial for protein synthesis within mitochondria, leading to a deficiency in ATP production [20].

Neuropathy, Ataxia, and Retinitis Pigmentosa (NARP) Syndrome represents a more specific OXPHOS disruption, driven by mutations in the ATP6 gene, affecting Complex V. Mutations like m.8993T > G disrupt the enzyme’s function, leading to impaired ATP synthesis [21], which is especially harmful to tissues with high energy demands, such as the nervous system and retina. The severity of symptoms in NARP depends on the heteroplasmy level (the proportion of mutated to normal mtDNA) and the tissue distribution of the mutated DNA [21,22].

Expanding the paradigm to include disorders with indirect effects on the ETC, Friedreich’s Ataxia (FRDA) highlights the significance of mitochondrial iron-sulfur cluster biogenesis. It is the most common hereditary ataxia, inherited in an autosomal recessive manner, with a prevalence of about 1 in 50,000. It is a disorder that although is primarily nDNA-related, affects mitochondrial iron-sulfur cluster biogenesis, impairing Complex I, II, and III function. FRDA is primarily due to a GAA trinucleotide repeat expansion in the FXN gene, which results in reduced expression of frataxin, a mitochondrial protein involved in iron-sulfur cluster assembly, iron homeostasis, and oxidative stress regulation. The deficiency of frataxin disrupts mitochondrial respiration and leads to iron accumulation, increasing oxidative damage [23].

A similar convergence occurs in Leber’s Hereditary Optic Neuropathy (LHON), which is caused by mtDNA mutations—11778G > A (ND4), 14484T > C (ND6), and 3460G > A (ND1)—that disrupt Complex I of the mitochondrial respiratory chain, causing oxidative stress. Retinal ganglion cells (RGCs), with their high energy demands and reliance on mitochondrial function, are particularly vulnerable to these disruptions, which underlies the hallmark symptoms of LHON- subacute bilateral vision loss due to optic atrophy, often manifesting in young males. Incomplete penetrance (~50% in males, ~10% in females) suggests the involvement of additional factors. Mitochondrial haplogroups, such as haplogroup J, may further modulate disease expression. The striking male predominance (3:1 to 8:1) remains incompletely understood, possibly involving hormonal or X-linked mechanisms [24,25].

Lafora Disease (LD), though classically defined as a glycogen storage disorder, further broadens the scope of OCPHOS-linked neurodegeneration. It is a progressive neurodegenerative disorder caused by autosomal-recessive mutations in the EPM2A and EPM2B genes. EPM2A encodes laforin, a glycogen phosphatase, while EPM2B encodes malin, an E3 ubiquitin ligase. Together, these proteins regulate glycogen metabolism by preventing the accumulation of hyperphosphorylated, insoluble glycogen aggregates called Lafora Bodies (LBs). Mutations in EPM2A or EPM2B disrupt glycogen dephosphorylation and ubiquitination pathways, leading to LB accumulation in neurons and other tissues. These aggregates impair cellular processes by disrupting mitochondrial dynamics, increasing oxidative stress, and impairing the OXPHOS pathway. The accumulation of Lafora contributes to neuronal dysfunction, synaptic impairment, astrocytic dysfunction, and loss of inhibitory interneurons which exacerbate neurodegeneration [26].

### 2.2. Pyruvate Metabolism Pathway

The pyruvate metabolism pathway links glycolysis to the tricarboxylic acid (TCA) cycle by converting pyruvate into acetyl-CoA via the pyruvate dehydrogenase complex (PDH). Defects in this pathway reduce substrate availability for ATP production. The disorders mentioned below, despite their distinct genetic etiologies and primary biochemical mechanisms, share a common pathophysiological theme: they each disrupt the critical metabolic checkpoint at the intersection of glycolysis, pyruvate oxidation, and TCA cycle entry, resulting in profound mitochondrial energy failure, lactate accumulation and selective neurodegeneration, especially in high demand.

In Congenital Lactic Acidosis (CLA), the disruption of the metabolic pathway is direct and is caused by mutations affecting mitochondrial energy metabolism, particularly in genes involved in pyruvate oxidation, such as PDHA1 (which encodes the E1-alpha subunit of the pyruvate dehydrogenase complex) and PDHX. These mutations impair the conversion of pyruvate to acetyl-CoA, leading to the accumulation of lactic acid. While CLA is most often associated with primary mitochondrial disorders—which can involve mutations in any of approximately 1500 mitochondrial or nuclear genes—some forms of lactic acidosis may also arise secondary to disorders like Glycogen Storage Disease Type IXa (PHKA2 mutations). CLA-related mutations can affect enzymes of the pyruvate dehydrogenase complex, the TCA cycle, and components of the ETC. Other genetic contributors include ACAD9, BCS1L, DGUOK, COQ2, FOXRED1, FARS2, GFM1, MRPS22, PDSS1, TMEM70, TRMU, TSFM, DLD, and SLC19A3—many of which are involved in respiratory chain function or multiple enzyme complex assembly. Because impaired energy production can affect virtually any organ or tissue, the clinical presentation is highly variable, and the diagnostic workup is often complex [27,28].

In contrast, Myoneurogastroencephalopathy (MNGIE), which is an autosomal recessive disease caused by mutations in the TYMP gene, involves an indirect but progressive impairment of the pyruvate metabolism pathway. This blockage leads to complete mitochondrial failure due to a progressive acquisition of secondary mtDNA mutations and mtDNA depletion affecting pyruvate metabolism. The disease is characterized by neurological and gastrointestinal symptoms, progressive degeneration and it leads to death at an average age of 37.6 years [29].

Lastly, on the other hand, Glutaric Acidemia Type I (GA-I) disrupts the pyruvate-to-TCA link via metabolite-mediated inhibition. It is an autosomal recessive disorder caused by mutations in the glutaryl-CoA dehydrogenase (GCDH) gene, as we can see in Table 1. GCDH is essential for the catabolism of lysine, hydroxylysine, and tryptophan. It catalyzes the oxidative decarboxylation of glutaryl-CoA to crotonyl-CoA in the mitochondrial matrix. Mutations in the GCDH gene result in deficient enzymatic activity, leading to the accumulation of glutaric acid, 3-hydroxyglutaric acid, and glutarylcarnitine in tissues and fluids. These metabolites disrupt cerebral energy metabolism, interfering with mitochondrial function and impairing the pyruvate metabolism pathway. In GA-I, accumulated metabolites inhibit pyruvate dehydrogenase, reducing acetyl-CoA production and TCA cycle flux. This energy deficit, compounded by oxidative stress, results in striatal vulnerability and neurodegeneration during encephalopathic crises [30,31].

**Table 1 cimb-47-00504-t001:** Mitochondrial disorders categorized by affected pathways such as the OXPHOS and pyruvate metabolism pathway in combination with their neurological features and inheritance patterns.

Mitochondrial Disorder	Gene(s)	Clinical Features	Inheritance	References
Leigh Syndrome	SURF1	Hypotonia, dystonia, hypopnea, dysphagia, epilepsy, failure to thrive, encephalopathy, basal ganglia and brainstem lesions	AR	[12]
AHS	POLG	Intractable epilepsy, psychomotor regression, liver disease	AR	[15]
MELAS	MT-TL1/MT-TF/MT-TV/MT-TQ	Stroke—like episodes, deafness, Diabetes Melitus, pigmented retinopathy, cardiomyopathy, cerebellar ataxia, seizures, encephalopathy, lactic acidosis, mitochondrial myopathy	M	[19]
NARP	MT-ATP6	Sensorimotor neuropathy, ataxia, pigmentary retinopathy, seizures, learning disability, dementia, proximal neurogenic muscle weakness, basal ganglia abnormalities	M	[21]
LHON	11778G > A (ND4), 14484T > C (ND6), 3460G > A (ND1)	Subacute bilateral vision loss due to optic atrophy and rarely tremors, dystonia, MS-like symptoms	M (mutations in the mtDNA)	[24]
LD	EPM2A or EPM2B	Asymptomatic until adolescence, patients undergo first insidious then rapid progressive myoclonus epilepsy toward a vegetative state and death within a decade. Neurodegenerative due to lysosomal dysfunction	AR	[26]
KSS	Mitochondrial tRNA tyrosine gene	Ophthalmoplegia, ptosis, progressive nature, muscle weakness beyond the eyes, fatigue, droopy mouth, cardiomyopathy	M	[18]
MERRF	MT-TK/MT-TF/MT-TL1/MT-TP	Progressive myoclonic epilepsy, ataxia, weakness, retinopathy, sensorineural hearing loss, lactic acidosis, lipomata, spasticity, cardiac function defects	M	[20]
FRDA	FXN	Iron accumulation, increasing oxidative damage	AR	[23]
CLA	PDHA1	Progressive neuromuscular weakness, accumulation of lactate in the blood (acidosis), urine and/or CSF	X, AR	[28]
MNGIE	TYMP	Peripheral neuropathy, leukoencephalopathy, ophthalmoplegia, ptosis, cachexia, gastrointestinal and dysmotility, progressively degenerative, leads to death at an average age of 37.6 years.	AR	[29]
GA -I	GCDH	Neurodegeneration due to encephalopathic crises, striatal vulnerability	AR	[31]

M: maternal, AR: autosomal recessive, X: X-linked.

### 2.3. Fatty Acid β-Oxidation Pathway

Fatty acid β-oxidation breaks down long-chain fatty acids into acetyl-CoA, providing substrates for the TCA cycle and subsequent ATP production through OXPHOS. This pathway is vital for energy generation, particularly during fasting.

Sengers Syndrome exemplifies a lipid-linked mitochondrial disorder with indirect effects on fatty acid oxidation. It is associated with mutations in acylglycerol kinase (AGK) [32], disrupting mitochondrial lipid metabolism and indirectly impairing fatty acid oxidation and OXPHOS. AGK mutations disrupt phospholipid synthesis and mitochondrial function, impairing ATP production and leading to oxidative damage. There are two forms of Sengers syndrome: a severe neonatal form that is typically fatal and a milder variant with a longer lifespan [33,34].

Glutaric Acidemia Type II, also known as Multiple Acyl-CoA Dehydrogenase Deficiency (MADD), represents a more direct failure of fatty acid oxidation affecting mitochondrial energy metabolism. It is an autosomal recessive disorder caused by mutations in genes encoding components of the mitochondrial electron transfer flavoprotein (ETF) complex. The key genetic contributors are Electron Transfer Flavoprotein Alpha Subunit (ETFA)**,** Electron Transfer Flavoprotein Beta Subunit (ETFB) and Electron Transfer Flavoprotein-Ubiquinone Oxidoreductase (ETFDH). ETF and ETFDH are crucial in transferring electrons from flavoprotein dehydrogenases (short-, medium-, and long-chain acyl-CoA dehydrogenases) to ubiquinone in the mitochondrial respiratory chain. Mutations in these genes disrupt β-oxidation of fatty acids, amino acid metabolism, and choline metabolism leading to the accumulation of toxic lipid intermediates in organs. The symptoms vary depending on the age of onset; neonatal or late [31,35,36].

In contrast, Andersen Disease, or Glycogen Storage Disease Type IV (GSD IV) and its adult-onset counterpart Adult Polyglucosan Body Disease (APBD), are autosomal recessive disorders that affect mitochondrial function indirectly through mutations in the GBE1 gene, which encodes the glycogen branching enzyme (GBE). This enzyme is essential for forming α-1,6 branch points in glycogen during synthesis. Mutations in GBE1 lead to defective glycogen branching, resulting in the accumulation of insoluble, linear glycogen molecules known as polyglucosan bodies. These structures disrupt cellular homeostasis, impair mitochondrial function, and interfere with pyruvate metabolism. Clinically, Andersen Disease presents as a spectrum, including impairments in energy-intensive tissues and the patients are categorized into hepatic and neuromuscular subtypes [37].

### 2.4. Amino Acid Metabolism and Phospholipid Remodeling Pathways

In this section, the mitochondrial disorders are caused by mutations affecting amino acid metabolism and phospholipid remodeling, contributing to dysregulation of intermediates to the TCA cycle and compromising the maintenance of membrane integrity.

Megdel Syndrome caused by mutations in *SERAC1*, demonstrates the direct impact of disrupted phospholipid remodeling on mitochondrial function. Mutations in the SERAC1 gene, which plays a role in phospholipid remodeling at the interface between the mitochondria and endoplasmic reticulum, are the underlying cause [38,39].

Fragile X Syndrome connects to this pathway through more mediated, regulatory mechanisms. The genetic pathway of Fragile X Syndrome (FXS) is centered on mutations in the FMR1 gene located on the X chromosome. This gene typically contains a CGG trinucleotide repeat, which expands abnormally in FXS patients. Full mutations (≥200 repeats) lead to hypermethylation, silencing FMR1 transcription and halting the production of fragile X mental retardation protein (FMRP), a critical regulator of mRNA translation at synapses. The absence of FMRP disrupts synaptic plasticity and neuronal development. FMRP influences both amino acid metabolism and phospholipid remodeling. It regulates proteins involved in the mTOR signaling pathway, crucial for amino acid sensing and downstream metabolic pathways. Dysregulation of mTOR activity in FXS disrupts amino acid utilization, contributing to synaptic abnormalities. In phospholipid remodeling, FMRP’s absence affects the stability of mRNAs encoding enzymes in lipid metabolism. This impacts synaptic membrane composition, impairing neural signaling and plasticity contributing to the neurodevelopmental deficits characteristic of FXS [40,41].

Ataxia Telangiectasia (A-T) adds a further layer to this metabolic-neurological interface. Although caused by mutations in the ATM gene—primarily associated with DNA damage response—ATM also participates in mitochondrial quality control and redox regulation. More specifically, ATM gene encodes a serine/threonine protein kinase essential for DNA double-strand break repair, cell cycle regulation, and oxidative stress responses. The ATM protein primarily functions in the DNA damage response pathway, where it phosphorylates key substrates involved in DNA repair, chromatin remodeling, and cell cycle checkpoints. However, ATM also plays a crucial role in maintaining mitochondrial homeostasis and responding to oxidative stress. A portion of ATM is localized to peroxisomes and mitochondria, where it regulates ROS metabolism, mtDNA repair, and mitochondrial quality control mechanisms. Loss of ATM function leads to chronic oxidative stress, mitochondrial dysfunction, and increased ROS production, contributing to neuronal degeneration due to loss of Purkinje cells, immune system impairment, metabolic dysregulation, and increased cancer susceptibility [42,43].

Ornithine Transcarbamylase Deficiency (OTCD) is an X-linked urea cycle disorder that further illustrates how disruptions in amino acid metabolism affect mitochondrial and cellular function. It is caused by pathogenic variants in the OTC gene. The OTC enzyme is crucial for converting carbamoyl phosphate and ornithine into citrulline in the urea cycle, a metabolic pathway responsible for removing excess nitrogen from the body by synthesizing urea. In OTCD, deficient OTC activity prevents the formation of citrulline from carbamoyl phosphate and ornithine. This results in the accumulation of ammonia and upstream nitrogenous compounds. Excess ammonia is detoxified by glutamine synthetase, leading to elevated glutamine and decreased glutamate levels. Glutamate, a key neurotransmitter and precursor for gamma-aminobutyric acid (GABA), is depleted, contributing to neurotoxicity and encephalopathy. The buildup of carbamoyl phosphate in OTCD shunts it into pyrimidine biosynthesis, leading to increased orotic acid levels. Disrupted pyrimidine metabolism affects nucleotide pools, impairing DNA and RNA synthesis required for cellular repair and proliferation. Arginine depletion reduces the synthesis of polyamines, which are crucial for stabilizing phospholipid membranes. Additionally, disturbances in amino acid metabolism impact phosphatidylcholine and sphingomyelin remodeling, essential for maintaining membrane integrity and cell signaling. Arginine deficiency disrupts systemic nitric oxide signaling, contributing to vascular and metabolic abnormalities [44,45].

### 2.5. Multi-System Pathways

These diseases are often classified as multi-system disorders because they can affect several interconnected pathways within the mitochondria, leading to widespread impacts on cellular and organ function.

Pearson Syndrome serves as a classic example of a mtDNA-related multisystem disorder that results from mtDNA deletions, impairing hematopoiesis and systemic energy metabolism [46]. It is typically caused by large-scale deletions or rearrangements in mtDNA. These deletions impair the function of genes involved in mitochondrial energy production. The condition usually arises from de novo mutations during early embryonic development, though it can also result from inherited mutations at varying levels of severity [47].

Fahr’s Syndrome, while primarily known for abnormal intracranial calcifications, is increasingly recognized for its possible mitochondrial involvement. Although direct mitochondrial mutations have not been consistently identified, several causative genes—*SLC20A2*, *PDGFB*, *PDGFRB*, and *XPR1*—play critical roles in phosphate transport and regulation. The first, SLC20A2, is responsible for about 40% of cases [48]. Mutations in platelet-derived growth factor subunit B (PDGFB) and platelet-derived growth factor receptor B (PDGFRB) account for approximately 11% and 2% of cases, respectively [49,50]. Additionally, mutations in the xenotropic and polytropic retrovirus receptor (XPR1) gene have been identified in about 2% of patients. In approximately half of the cases, no mutations are detected, leaving the cause unknown. The SLC20A2 gene is located on chromosome 8p11.2 and encodes the type III sodium-dependent inorganic phosphate transporter 2 (PiT2), a crucial transmembrane Na^+^/Pi cotransporter that maintains phosphate (Pi) homeostasis necessary for ATP synthesis [51].

Rett Syndrome illustrates how mutations in nuclear genes can profoundly affect mitochondrial health through transcriptional regulation. As we can see, also, in Table 2, it is caused by mutations in the X-linked MECP2 gene, which encodes methyl-CpG-binding protein 2, a crucial regulator of gene expression. MECP2 functions by binding to methylated DNA and recruiting chromatin remodeling complexes to regulate transcription [52]. Loss of MECP2 function leads to widespread transcriptional dysregulation, affecting genes involved in neuronal maturation, synaptic function, and metabolic processes. One major consequence of MECP2 deficiency is mitochondrial dysfunction, particularly in mtDNA maintenance. Rett syndrome patients exhibit abnormalities in mitochondrial structure, impaired ETC function, increased oxidative stress, and altered mtDNA copy number. These mitochondrial defects contribute to the neurodegeneration seen in Rett syndrome, as neurons are highly dependent on mitochondrial energy production. Systemically, the disorder affects multiple organ systems, including the CNS, lead to metabolic dysfunctions, cardiovascular abnormalities, breathing irregularities, and gastrointestinal issues, further highlighting the systemic impact of MECP2 mutations [53,54].

**Table 2 cimb-47-00504-t002:** Mitochondrial disorders associated with defects in the fatty acid β-oxidation pathway, amino acid metabolism, phospholipid remodeling, and multi-system pathways in combination with their inheritance modes and clinical features.

Mitochondrial Disorder	Gene(s)	Clinical Features	Inheritance	References
Sengers Syndrome	AGK	Congenital Cataracts, Hypertrophic Cardiomyopathy, Lactic Acidosis, Myopathy, Exercise Intolerance	AR	[34]
GA-II	ETFA or ETFB or ETFDH	Hepatomegaly, non-ketotic hypoglycemia, metabolic acidosis, hypotonia, and in neonatal onset cardiomyopathy	AR	[31]
GSD- IV	GBE1 (severe mutations)	Profound skeletal muscle weakness, respiratory failure, death during early infancy, cardiomyopathy	AR	[37]
APBD	GBE1 (Milder mutations)	Neurogenic bladder dysfunction, spastic paraplegia, axonal neuropathy, cognitive symptoms, dementia	AR	[37]
Megdel Syndrome	SERAC1	Sensorineural hearing loss, encephalopathy, failure to thrive, hypotonia, psychomotor delay, dystonia, spasticity, hypoglycemia, hepatopathy, lactic acidosis	AR	[39]
Fragile X Syndrome	FMR1	Depends on whether premutation (tremor, ataxia)/full mutation (postpubertal macroorchidism, long face, large, everted ears, autism, hypermobile joints)	X- Dominant	[41]
A-T	ATM	Progressive cerebellar ataxia, oculocutaneous telangiectasia, variable immunodeficiency, radiosensitivity, susceptibility to malignancies, increased metabolic diseases	AR	[43]
OTCD	OTC	All due to hyperammonemia: somnolence, lethargy, coma, acute encephalopathy, frequent migraine headaches, seizures, tremor, ataxia, dysarthria, stroke-like episodes, transient visual loss, chorea, and protracted visual loss), psychiatric and gastrointestinal clinical manifestations	X	[44]
Pearson Syndrome	Large scale deletions in mtDNA	Sideroblastic anemia of childhood associated with exocrine or endocrine pancreatic dysfunction, pancytopenia, renal tubulopathy	M	[47]
Fahr’s Syndrome	SLC20A2, PDGFB, PDGFRB, XPR1	Parkinsonism, dystonia, chorea, dementia, mood changes, seizures, speech and swallowing difficulties	AD, (in some cases AR)	[55]
Rett Syndrome	MECP2	Metabolic dysfunctions, cardiovascular abnormalities, breathing irregularities, and gastrointestinal issues	X	[54]

M: maternal, AR: autosomal recessive, X: X-linked, AD: autosomal dominant.

### 2.6. Neurodegeneration with Brain Iron Accumulation

Neurodegeneration with Brain Iron Accumulation is a group of genetically inherited disorders characterized by excessive iron deposition in the basal ganglia, particularly the globus pallidus and substantia nigra. These disorders share common underlying mechanisms involving iron dysregulation, mitochondrial dysfunction, lipid metabolism abnormalities, and impaired autophagy. The accumulation of iron leads to oxidative stress, protein aggregation, and cellular damage, contributing to progressive neurodegeneration. Although the specific genetic mutations vary, as we can see, also, on Table 3, they converge on pathways essential for maintaining neuronal function, metal homeostasis, and cellular energy balance [55].

Pantothenate kinase-associated neurodegeneration (PKAN) is an autosomal recessive disorder that demonstrates the link between CoA metabolism and mitochondrial iron homeostasis. It is caused by mutations in the PANK2 gene, which encodes pantothenate kinase 2, a key enzyme in coenzyme A (CoA) biosynthesis. The PANK2 enzyme is localized in the mitochondrial intermembrane space, where it regulates mitochondrial function by facilitating fatty acid metabolism, iron-sulfur cluster formation, and energy production. The loss of PANK2 function leads to disrupted CoA biosynthesis, resulting in defective mitochondrial lipid metabolism, accumulation of toxic cysteine-containing compounds, and oxidative stress. These defects impair iron homeostasis, leading to excessive iron accumulation in the globus pallidus, a hallmark of PKAN. Additionally, mitochondrial acyl carrier protein (mtACP), which depends on CoA for activation, is downregulated, leading to impaired ETC function and increased oxidative damage. The systemic effects of PKAN include progressive neurodegeneration, dystonia, parkinsonism, visual impairment due to retinal degeneration, speech disorders, and psychiatric symptoms [56,57].

Closely related is CoA synthase protein-associated neurodegeneration (CoPAN), caused by mutations in the COASY gene, which is downstream of PANK2 in the CoAA biosynthetic pathways and encodes an essential enzyme involved in CoA biosynthesis, a critical molecule for mitochondrial function and cellular metabolism. Loss of COASY function leads to disrupted mitochondrial bioenergetics, oxidative stress, and iron dyshomeostasis, particularly in the brain [58]. Studies in models such as yeast, zebrafish, and mice have demonstrated that COASY deficiency results in mitochondrial dysfunction, including impaired respiratory chain activity and reduced ATP production [59]. The disorder primarily affects the nervous system, leading to progressive motor dysfunction, dystonia, and cognitive decline, with varying severity depending on the specific mutation [58].

In contrast, Phospholipase A2-associated neurodegeneration (PLAN) bridge mitochondrial dysfunction with impaired phospholipid metabolism and autophagy. It is caused by mutations in the PLA2G6 gene, which encodes phospholipase A2 group VI, an enzyme responsible for catalyzing the release of free fatty acids from phospholipids. Mutations in PLA2G6 lead to phospholipid dyshomeostasis affecting mitochondrial membranes. Neuropathological studies in patients with PLAN have revealed widespread neuroaxonal dystrophy, α-synuclein pathology, and tau accumulation, implicating this disorder in broader neurodegenerative mechanisms. Brain iron accumulation is also a key feature of PLAN, particularly in the basal ganglia, where excess iron exacerbates oxidative stress and neuronal damage. Clinically, PLAN manifests as a spectrum of disorders, ranging from infantile neuroaxonal dystrophy with early-onset neurodevelopmental regression to later-onset dystonia-parkinsonism. Additionally, peripheral nerve degeneration and sensorimotor dysfunction have been observed, further highlighting the systemic impact of PLA2G6 mutations on both the central and peripheral nervous systems [60,61].

In the same context, Mitochondrial Membrane Protein-Associated Neurodegeneration (MPAN), also, reflects the mitochondrial dysregulation due to autophagy and abnormal phospholipid metabolism. It is caused by autosomal-recessive mutations in the C19orf12 gene. The C19orf12 protein is located on the outer mitochondrial membrane, mitochondria-associated membranes, and the ER. Mutations in C19orf12 disrupt lipid homeostasis and fatty acid biogenesis, leading to defective mitochondrial function, impaired phospholipid transport, and altered myelin production. Dysfunctional autophagy causes accumulation of damaged mitochondria, exacerbating oxidative stress, apoptosis, and cellular toxicity. Neuropathological features include widespread Lewy body pathology, axonal spheroids, neuronal loss, and iron-containing deposits, clinically presenting with neurological and neuropsychiatric symptoms. Mutations like p.Thr11Met are associated with adult-onset disease, while p.Gly69ArgfsX10 mutations result in early-onset disease with optic atrophy and spastic paraparesis [62,63].

Expanding on the theme of impaired organelle quality control, Beta-Propeller Protein-Associated Neurodegeneration (BPAN) highlights the downstream effects of failed autophagic clearance. It is caused by mutations in the WDR45 gene, located on the X chromosome, which encodes a WD-repeat β-propeller protein involved in autophagy regulation. WDR45 plays a key role in cellular homeostasis by facilitating the formation and maturation of autophagosomes, which are essential for the degradation and recycling of cellular components [64]. Mutations in WDR45 lead to impaired autophagic flux, resulting in the accumulation of damaged mitochondria that contributes to iron dysregulation in the brain, and increased oxidative stress [65]. Clinically, BPAN is characterized by early-onset developmental delay, intellectual disability, seizures, and progressive motor and cognitive decline in adulthood, often accompanied by parkinsonism and dystonia [64].

In this interconnected framework, Kufor-Rakeb Syndrome (KRS) serves as a bridge between lysosomal dysfunction and mitochondrial damage. It is an autosomal recessive disorder caused by mutations in the ATP13A2 gene, which encodes a P-type ATPase involved in lysosomal polyamine transport. ATP13A2 plays a crucial role in maintaining cellular proteostasis by regulating lysosomal function, autophagy, and mitochondrial quality control. Loss of ATP13A2 function leads to impaired lysosomal acidification, defective protein degradation, and mitochondrial dysfunction. ATP13A2 mutations contribute to iron accumulation in the brain, particularly in the basal ganglia, exacerbating neuronal damage through oxidative stress and ferroptosis. KRS primarily affects the nervous system, causing juvenile-onset parkinsonism, cognitive impairment, myoclonus, dystonia, and pyramidal signs and psychiatric symptoms [66,67].

On the contrary, Neuroferritinopathy and Aceruloplasminemia represent disorders where the primary defect lies in iron-handling proteins, yet both culminate in mitochondrial dysfunction. Firstly, Neuroferritinopathy is a neurodegenerative disorder caused by mutations in the FTL gene, which encodes the light chain subunit of ferritin, the primary intracellular iron storage protein. Mutant FTL disrupts ferritin’s ability to properly sequester iron, leading to an increase in the cellular iron pool and excessive production of ROS. This imbalance results in oxidative stress, protein aggregation, and mitochondrial dysfunction. Studies in transgenic mice and cell models expressing mutant FTL show widespread iron accumulation in the brain, particularly in the basal ganglia, along with altered gene expression related to iron metabolism, increased lipid peroxidation, and mtDNA damage. Brain iron accumulation in neuroferritinopathy is associated with cystic degeneration and progressive movement disorders, including dystonia, tremor, and parkinsonism. The disease can also affect peripheral tissues, as ferritin inclusions and iron overload have been observed in non-neuronal cells. The combination of impaired iron homeostasis, oxidative stress, and mitochondrial dysfunction plays a central role in the progression of neuroferritinopathy and its systemic effects [68,69].

Secondly, Aceruloplasminemia, which is a rare autosomal recessive disorder caused by mutations in the CP gene, which encodes ceruloplasmin, a ferroxidase enzyme essential for iron homeostasis. Ceruloplasmin exists in two forms: a secreted form primarily expressed in the liver and a GPI-linked form expressed in the brain. Mutations in CP, often deletions or insertions leading to frameshifts and premature stop codons, result in defective ceruloplasmin production, leading to ER stress, impaired copper incorporation, and reduced ferroxidase activity. The loss of ceruloplasmin function disrupts iron metabolism by destabilizing ferroportin, preventing proper iron efflux and causing intracellular iron accumulation, particularly in astrocytes. contributing to movement disorders, cognitive decline, psychiatric symptoms, diabetes mellitus due to pancreatic iron deposition and retinal degeneration [69].

Lastly, Woodhouse-Sakati Syndrome (WSS), though less understood, may serve as a connecting point between nuclear integrity, mitochondrial dysfunction, and iron accumulation. As we can see in Table 3, it is an autosomal recessive neurodegenerative disorder caused by mutations in the DCAF17 gene, also known as C2orf37. This gene encodes a nucleolar transmembrane protein involved in nuclear organization and cellular homeostasis, though its exact function remains unclear. Mutations in DCAF17 lead to disruption of the nuclear membrane, which affects key cellular processes such as transcriptional regulation, RNA processing, apoptosis, and cytoskeletal integrity. Cytoskeletal defects caused by the loss of DCAF17 function contribute to abnormal cellular structure, impaired intracellular transport, and neuronal dysfunction, which may underlie the progressive neurodegeneration observed in WSS. The iron accumulation in the brain leads to movement disorders and systemically WSS results in endocrine dysfunction, sensorineural hearing loss, intellectual disability, and dysmorphic facial features [70,71].

**Table 3 cimb-47-00504-t003:** Mitochondrial disorders associated with neurodegeneration and brain iron accumulation in combination with their clinical features and their respective modes of inheritance.

Mitochondrial Disorder	Gene(s)	Clinical Features	Inheritance	References
PKAN	PANK2	Dystonia, parkinsonism, spasticity, pigmentary retinopathy, acanthocytosis, neuropsychiatric features	AR	[57]
PLAN	PLA2G6	Psychomotor regression, ataxia, autism, dystonia, parkinsonism, optic atrophy	AR	[61]
Neuroferritinopathy	FLT	Adult- onset chorea or dystonia with subtle cognitive defects	AR	[69]
MPAN	C190rf12	Spasticity, dystonia, dementia, peripheral nerve involvement	AR	[63]
CoPAN	CoASY	Intellectual disability, dystonia, spasticity, behavioral problems	AR	[59]
BPAN	WDR45	Intellectual disability, little to no language, mixed seizure types, juvenile parkinsonism, autism	X	[65]
KRS	ATP13A2	Juvenile parkinsonism, dementia	AR	[67]
Aceruloplasminemia	CP	Adult- onset retinal degeneration, diabetes mellitus, chorea/dystonia/ataxia	AR	[70]
WSS	DCAF17	Movement disorders, endocrine dysfunction, sensorineural hearing loss, intellectual disability, dysmorphic facial features	AR	[71]

AR: autosomal recessive, X: X-linked.

## 3. Genetic and Metabolic Complexity of Mitochondrial Disorders in the Era of Precision Medicine

The expanding field of mitochondrial medicine highlights the intricate interplay between genetic mutations, metabolic disruptions, and the phenotypic variability observed across a broad spectrum of disorders. Given the complexity of mitochondrial diseases, traditional diagnostic and therapeutic paradigms often prove inadequate, frequently failing to address the underlying heterogeneity in disease mechanisms and patient presentations [72]. Precision medicine has emerged as a promising strategy to overcome these limitations by tailoring diagnosis, treatment, and management to each individual’s genetic background, biochemical profile, and clinical characteristics. Advances in genetic testing, including whole-exome and whole-genome sequencing [73], now allow for early and accurate identification of pathogenic variants such as MT-ND gene mutations in Leigh syndrome and POLG mutations in Alpers-Huttenlocher syndrome [74]. These diagnostic tools enhance the ability to predict disease trajectories and inform personalized therapeutic approaches. Targeted metabolic therapies, such as ketogenic diets for pyruvate dehydrogenase deficiency, exemplify how mechanism-based interventions can restore metabolic homeostasis and improve clinical outcomes, signaling a transition from symptomatic management to etiology- informed therapies.

Therapeutic innovation in mitochondrial medicine is rapidly advancing, driven by developments in gene- and pathway-specific interventions. Gene-editing technologies, especially CRISPR-based systems, offer potential for correcting pathogenic mutations at the DNA level [75]. However, their application to the mitochondrial genome is constrained by current limitations in mitochondrial RNA delivery technologies. To address this, CRISPR-free mitochondrial base editing platforms are under active preclinical investigation and may soon provide more feasible tools for modifying mtDNA with greater specificity and fewer delivery barriers. In parallel, recombinant adeno-associated virus (rAAV) vectors are currently being tested in clinical trials across the United States, Europe, and China for the treatment of nuclear gene-related mitochondrial disorders [76]. While these trials represent a major step forward, somatic mitochondrial diseases are not currently amenable to correction through mitochondrial replacement therapy (MRT), which remains limited to preventing the maternal transmission of mtDNA mutations. MRT involves the transfer of the nuclear genome from a mother carrying a pathogenic mtDNA mutation into an enucleated donor oocyte with healthy mitochondria [77]. Despite these advancements, several challenges remain, including the efficient delivery of therapeutics to affected tissues, potential off-target effects, and the difficulty of crossing the blood-brain barrier in conditions with neurological involvement. Additionally, heteroplasmy, where both mutated and wild-type mtDNA coexist within cells, introduces unpredictable variation in disease severity and therapeutic response [78].

To fully realize the potential of precision medicine in mitochondrial disorders, deeper integration of molecular and clinical insights is essential. This review proposes a pathway-based framework for organizing disorders with neurological involvement according to the specific mitochondrial functions impaired. Rather than offering rigid disease classifications, this conceptual model groups disorders by the metabolic pathways primarily affected. These include oxidative phosphorylation, pyruvate metabolism, fatty acid β-oxidation, amino acid metabolism, phospholipid remodeling, multi-systemic mitochondrial regulation, and mechanisms related to neurodegeneration with brain iron accumulation. This approach enhances our ability to link genotypes to phenotypes and to uncover therapeutic targets by revealing biochemical dependencies unique to each disease subtype [79]. The growing field of pharmacogenomics also supports this individualized model by enabling the personalization of drug regimens based on genetic variations in drug metabolism, mitochondrial transport, and membrane integrity [80]. Prospectively, the integration of multi-omics technologies such as transcriptomics, proteomics, and metabolomics, combined with systems biology and machine learning, offers the opportunity to establish computational prediction for disease progression and treatment response. Achieving clinical implementation will require investment in mitochondrial-specific delivery systems, scalable gene-editing tools, and large, collaborative cohort studies to validate these pathway-based approaches across diverse populations.

## 4. Conclusions

The integration of genetic evidence with a pathway-based understanding of mitochondrial dysfunction offers a powerful framework for advancing both the diagnosis and treatment of disorders with neurological features. Mitochondrial diseases, despite their clinical and genetic diversity, share a unifying theme of disrupted metabolic homeostasis across key pathways such as oxidative phosphorylation, pyruvate metabolism, fatty acid β-oxidation, amino acid metabolism, phospholipid remodeling, and neurodegeneration with brain iron accumulation. This classification enables a more mechanistic interpretation of disease phenotypes, supporting focused investigation and evidence-based clinical decision-making. The dual genetic origin of mitochondrial proteins, involving both nuclear and mitochondrial DNA, adds further complexity to diagnostic strategies and therapeutic development. Nevertheless, precision medicine approaches, supported by advances in genomic technologies such as whole-exome sequencing, pharmacogenomic profiling, and molecular pathway mapping, are beginning to narrow the gap. Innovative therapeutic modalities, including mitochondrial gene editing, metabolic reprogramming, and mitochondrial replacement therapy, are paving the way toward personalized therapeutic approaches. However, significant challenges remain, including the biological variability introduced by heteroplasmy, the complexity of lactylation-driven epigenetic dysregulation, and the tissue-specific manifestations of mitochondrial dysfunction [81]. Lactylation, an emerging post-translational modification linked to altered metabolic states, has been implicated in modulating gene expression and inflammatory responses, thereby contributing to the progression of neurological pathology in mitochondrial diseases [82]. Moreover, therapeutic delivery to affected neural tissues remains a major obstacle. Promisingly, recent findings have highlighted the ARRB1/S100A9 signaling axis as a potential therapeutic target capable of modulating mitochondrial stress responses and neuroinflammation [83]. Future progress will depend on sustained efforts to unravel the molecular intricacies of mitochondrial impairment—including these novel regulatory layers—and to translate such insights into effective, patient-centered treatments that improve both clinical outcomes and quality of life.

## Figures and Tables

**Figure 1 cimb-47-00504-f001:**
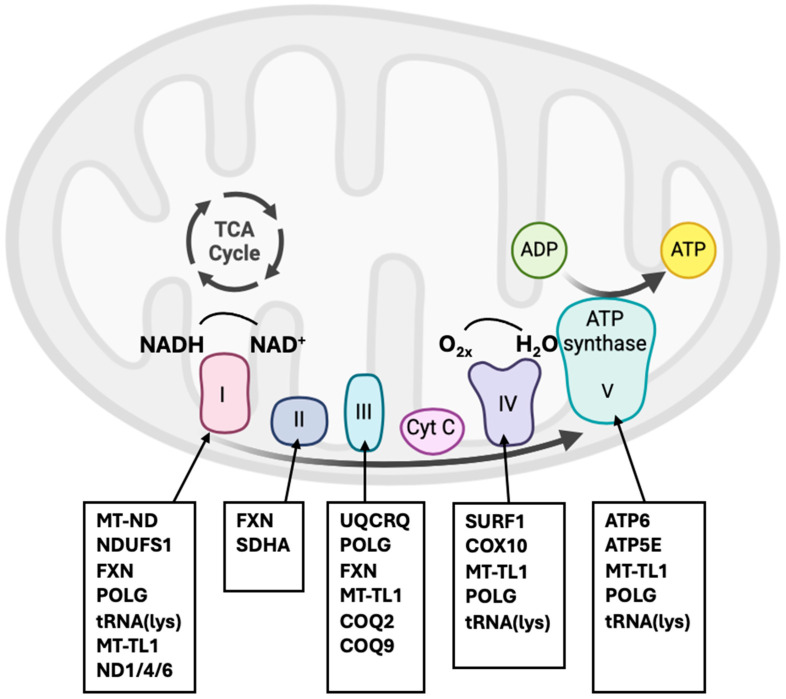
Overview of the oxidative phosphorylation (OXPHOS) pathway within the mitochondrial inner membrane, highlighting the five multi-subunit enzyme complexes (I–V) responsible for ATP production. Electrons derived from NADH and FADH_2_ are transferred through complexes I–IV, generating a proton gradient that drives ATP synthesis at complex V (ATP synthase). Molecular oxygen (O_2_) serves as the terminal electron acceptor, forming water (H_2_O) at complex IV. Below each complex, nuclear and mitochondrial genes most commonly associated with mitochondrial disorders are listed. Mutations in these genes disrupt specific steps in the OXPHOS pathway, impairing ATP production and leading to energy deficits that particularly affect high-demand tissues such as the brain. Created with BioRender.com.

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
