# Peer review of "Mapping Disorders with Neurological Features Through Mitochondrial Impairment Pathways: Insights from Genetic Evidence"

_cimb, 2025, doi:10.3390/cimb47070504_

Round 1

Reviewer 1 Report

Comments and Suggestions for Authors

Journal: CIMB (ISSN 1467-3045)

Manuscript ID: cimb-3677534

Type: Review

Title: Mapping Disorders with Neurological Features through Mitochondrial Impairment Pathways: Insights from Genetic Evidence

This review titled “Mapping Disorders with Neurological Features through Mitochondrial Impairment Pathways: Insights from Genetic Evidence” by Anna Makridou et al.; categorizes neurological disorders caused by mitochondrial dysfunction using a pathway-based approach. It classifies these disorders based on the specific mitochondrial processes that are impaired—such as oxidative phosphorylation, pyruvate metabolism, fatty acid oxidation, and others—and associates them with genetic mutations, clinical features, and inheritance patterns. By integrating genetic and molecular data, the review hopes to improve understanding of disease mechanisms and support the development of precision diagnostics and targeted therapies.

1: While the introduction provides a good explanation of mitochondrial dysfunction, it does not highlight the importance of genetic mutations, inheritance patterns, and recent genomic discoveries in driving neurological manifestations. Include a paragraph in introduction section describing how mutations in mitochondrial and nuclear genes are linked to specific neurological phenotypes.

2: lines 87-94: The word "mapping" is central to the title, but there is no definition of mapping in this review. Is it visual, tabular, or analytical mapping? How are disorders associated with pathways?

3: Conditions such as MELAS, Leigh syndrome, and fragile X syndrome are named but not associated with specific pathways or genes. This reduces the clarity of how the review will link disorders to mitochondrial dysfunction.

4: Should "amino acid catabolism" be changed to "amino acid metabolism" to align with subsequent pathway classifications?

5: Line 69: Aside from cytochrome c, are there any other pro-apoptotic factors that could be briefly mentioned (e.g., AIF, SMAC)?

6: Can the phrase "offer a structured perspective" be replaced by a more specific description of the review's contribution (for example, "introduces a novel gene-pathway-disease framework")?

7: The closing lines (93–95) are general and don’t highlight what the reader will gain. Finish with a strong, reader-focused statement such as:

“This review provides a structured, genetically informed map of mitochondrial pathways linked to neurological disorders, offering new avenues for diagnostics and targeted therapies.”

8: Briefly mention a figure or table that summarizes gene-pathway-disorder associations as a tool for readers (e.g., “Figure 1 visually summarizes gene-pathway relationships discussed in this review.”).

Author Response

Firstly, we would like to express our gratitude to Reviewer 1 for their insightful suggestions and valuable comments, which have significantly enhanced the overall integrity of our manuscript. Below you may find our responses to each point raised, respectively.

1.While the introduction provides a good explanation of mitochondrial dysfunction, it does not highlight the importance of genetic mutations, inheritance patterns, and recent genomic discoveries in driving neurological manifestations. Include a paragraph in the introduction section describing how mutations in mitochondrial and nuclear genes are linked to specific neurological phenotypes.

Response: Thank you for this important observation. We added a paragraph in the introduction (now lines 66-88) where we elaborated on the genotype–phenotype correlations, highlighting how both mitochondrial and nuclear gene mutations contribute to neurological presentations. This revision emphasizes recent discoveries and inheritance patterns relevant to mitochondrial disorders.

“A critical aspect of mitochondrial-related neurological disorders is the complex and often unpredictable relationship between genotype and phenotype, shaped by both the origin of the mutation—mitochondrial or nuclear—and its mode of inheritance. Mutations in mitochondrial DNA (mtDNA) typically follow maternal inheritance and can result in heteroplasmy, where mutant and wild type mtDNA coexist within cells. Heteroplasmy, the coexistence of normal and mutated mitochondrial DNA within cells, contributes to the unpredictable severity and distribution of symptoms across individuals and tissues. The proportion of mutant mtDNA can vary across tissues and over time, influencing disease severity, age of onset, and symptom heterogeneity[4,5]. In contrast, nuclear-encoded mitochondrial gene mutations adhere to Mendelian inheritance patterns—autosomal dominant, autosomal recessive, or X-linked—allowing clearer, though not always straightforward, predictions of disease risk and transmission. Importantly, identical mutations in either mtDNA or nDNA can give rise to divergent clinical outcomes depending on background genetic modifiers, tissue energy demands, and environmental influences. For example, biallelic POLG mutations can lead to a spectrum of phenotypes ranging from childhood-onset epilepsy to adult-onset ataxia or myopathy, demonstrating how nuclear gene defects can have pleiotropic effects[6]. Understanding these genotype–phenotype correlations and their inheritance frameworks is essential for accurate diagnosis, risk assessment, and genetic counseling, as well as for identifying patients who may benefit from emerging precision therapies. Furthermore, expanding genomic databases and functional annotation of variants continue to refine our ability to map specific genetic defects to distinct neurological presentations, paving the way for precision medicine in mitochondrial diseases.”

2. Lines 87–94: The word “mapping” is central to the title, but there is no definition of mapping in this review. Is it visual, tabular, or analytical mapping? How are disorders associated with pathways?

Response: Thank you for this observation and suggestion. "Mapping" refers primarily to analytical and tabular mapping. The associations between mitochondrial impairment pathways and neurological disorders are explored through genetic evidence, discussed in detail in the text and organized in tables to highlight key connections. While the paper does not include visual diagrams, it emphasizes conceptual mapping through analysis and structured presentation.

In line 95 we added a clarification analytical and tabular mapping”.

3. Conditions such as MELAS, Leigh syndrome, and fragile X syndrome are named but not associated with specific pathways or genes. This reduces the clarity of how the review will link disorders to mitochondrial dysfunction.

Response:  Thank you for your comment. You are absolutely right, Leigh syndrome is a genetically heterogeneous disorder associated with numerous genes affecting various aspects of mitochondrial function, including complex I, IV, V of the respiratory chain, pyruvate metabolism, and coenzyme Q10 biosynthesis, among others. Due to the extensive number of genes involved, we did not aim to exhaustively list each one. Instead, our focus was to highlight representative genetic mechanisms and common mitochondrial pathways disrupted in Leigh syndrome, which serve as a basis for understanding its pathogenesis. In the final version of the manuscript, we clarified this point to avoid confusion and included the most common gene and its pathway.

Line 149-152- “The most common gene causing Leigh syndrome is SURF1. Loss-of-function mutations in SURF1 impair complex IV activity, leading to reduced ATP production, increased lactate accumulation, and neuronal energy failure.”

We also carefully considered your comments regarding MELAS and fragile X syndrome. We would like to note that, in the current version of the manuscript, these conditions are already discussed in relation to their associated genes and mitochondrial pathways. Specifically, for MELAS, we describe mutations in mitochondrial tRNA genes (e.g., MT-TL1) and their impact on mitochondrial protein synthesis and oxidative phosphorylation. For Fragile X syndrome, we highlight the role of FMR1 and its effects on mitochondrial function. We have reviewed these sections to ensure clarity and emphasize these associations more explicitly where needed.

4. Should "amino acid catabolism" be changed to "amino acid metabolism" to align with subsequent pathway classifications?

Response: We agree and have updated the term to “amino acid metabolism” in all relevant sections (lines 116, 316, and 330) to maintain consistency across the manuscript.

5. Line 69: Aside from cytochrome c, are there any other pro-apoptotic factors that could be briefly mentioned (e.g., AIF, SMAC)?

Response: Thank you for your valuable suggestion. We included additional pro-apoptotic factors such as AIF and SMAC to provide a more comprehensive overview of mitochondrial involvement in apoptosis.

Line 122-126 “For instance, calcium overload can trigger the opening of the mitochondrial permeability transition pore (mPTP), releasing pro-apoptotic factors such as cytochrome c, as well as apoptosis-inducing factor (AIF), SMAC/DIABLO, and Endonuclease G, which together mediate both caspase-dependent and independent cell death pathways”

6. Can the phrase "offer a structured perspective" be replaced by a more specific description of the review's contribution (for example, "introduces a novel gene-pathway-disease framework")?

Response: Thank you for this suggestion. We rephrased the sentence to communicate better the aim of the review in lines 100-103. “This review provides a structured, genetically informed map of mitochondrial pathways linked to neurological disorders, offering new avenues for diagnostics and precision therapies, ultimately supporting advances in mechanistic understanding and precision medicine.”

7. The closing lines (93–95) are general and don’t highlight what the reader will gain. Finish with a strong, reader-focused statement such as:
“This review provides a structured, genetically informed map of mitochondrial pathways linked to neurological disorders, offering new avenues for diagnostics and targeted therapies.”

Response: We appreciate this recommendation and have revised the concluding sentences of the introduction (lines 100-103) accordingly. We retained the reference to precision medicine to emphasize clinical relevance while adopting the suggested phrasing to strengthen our message.

“This review provides a structured, genetically informed map of mitochondrial pathways linked to neurological disorders, offering new avenues for diagnostics and precision therapies, ultimately supporting advances in mechanistic understanding and precision medicine.”

8. Briefly mention a figure or table that summarizes gene-pathway-disorder associations as a tool for readers.

Response: We thank reviewer 1 for this thoughtful suggestion. We would like to note that the existing tables in the manuscript comprehensively present the gene-pathway-disorder associations in a structured and detailed format. As these tables already provide an integrated overview of the relevant information, we believe they effectively serve the purpose of guiding the reader through these associations. We hope the reviewer finds them sufficient in this regard.

Once again, we thank Reviewer 1 for the insightful and constructive feedback, which helped us improve the clarity, coherence, and scientific rigor of our manuscript.

Reviewer 2 Report

Comments and Suggestions for Authors

The manuscript lacks a crucial methodology section, which is mandatory for systematic reviews according to journal guidelines and PRISMA standards. While the content demonstrates significant expertise and provides valuable clinical insights, the absence of systematic methodology undermines the scientific rigor and precludes publication. The inclusion of a comprehensive methods section is essential.

Issues requiring attention:

  • The abstract is informative but lacks fluidity; sentences require better connectivity, and methodology must be incorporated.
  • A comprehensive grammatical review of the manuscript is required.
  • Tables need consistent formatting and standardization.
  • All research has limitations; therefore, please include a limitations section addressing the constraints of this study.
  • The manuscript would benefit from additional figures, particularly a comprehensive synthesis figure illustrating metabolic pathways and associated diseases.
  • Inconsistencies exist in reference formatting; this requires correction.
Comments on the Quality of English Language

A comprehensive grammatical review of the manuscript is required

Author Response

Firstly, we would like to express our gratitude to Reviewer 2 for their insightful suggestions and valuable comments, which have significantly enhanced the overall integrity of our manuscript. Below you may find our responses to each point raised, respectively.

Reviewer 2 Comment:
The manuscript lacks a crucial methodology section, which is mandatory for systematic reviews according to journal guidelines and PRISMA standards. While the content demonstrates significant expertise and provides valuable clinical insights, the absence of systematic methodology undermines the scientific rigor and precludes publication. The inclusion of a comprehensive methods section is essential.

Response:
Thank you for this important observation. We would like to clarify that our manuscript is intended as a literature review, not a systematic review. As such, it is not subject to PRISMA guidelines, and we did not follow a systematic review methodology. Our goal was to provide a broad synthesis of current findings and insights, supported by selected representative studies. In response to your comment, we have explicitly clarified the nature of the review in the revised manuscript (lines 94-95 and 100-103) to avoid any potential misunderstanding regarding its methodological framework: “To narrow the gap, the present narrative review proposes a pathway-oriented approach to analytical and tabular mapping disorders with neurological involvement, grounded in the specific mitochondrial processes affected.” and “This review provides a structured, genetically informed map of mitochondrial pathways linked to neurological disorders, offering new avenues for diagnostics and precision therapies, ultimately supporting advances in mechanistic understanding and precision medicine.”, respectively.

Reviewer 2 Comment:
The abstract is informative but lacks fluidity; sentences require better connectivity, and methodology must be incorporated.

Response:
Thank you for your suggestion. We have revised the abstract to improve the flow and connectivity between sentences. Additionally, we have clarified the nature of the review in the abstract, specifying that it is a narrative review to align with the structure and purpose of the manuscript. These additions can be found in lines 22-26 and 30-33 as follows: This narrative review provides a structured synthesis of current knowledge by classifying mitochondrial-related neurological disorders according to disrupted biochemical pathways, in order to clarify links between genetic mutations, metabolic impairments, and clinical phenotypes. More specifically, a pathway-oriented framework was adopted that organizes disorders based on the primary mitochondrial processes affected” and “A comprehensive table synthesizes genetic causes, inheritance patterns, and neurological manifestations across disorders. This approach offers a conceptual framework that connects molecular findings to clinical practice, supporting more precise diagnostic strategies and the development of targeted therapies.” respectively.

Reviewer 2 Comment:
A comprehensive grammatical review of the manuscript is required.

Response:
We appreciate this feedback. The manuscript has undergone a comprehensive grammatical and stylistic review by a native English speaker with academic writing expertise. A second round of thorough proofreading was conducted to enhance clarity, consistency, and overall language quality. All changes have been highlighted in yellow in the revised manuscript. Representative revisions include:

·        We replaced “a shift from symptom-based care to treatments grounded in molecular etiology” with “a transition from symptomatic management to etiology-informed therapies” (line 571)

·        We replaced “remains limited due to the technical challenge of importing guide RNAs into mitochondria” with “is constrained by current limitations in mitochondrial RNA delivery technologies” (line 576)

·        We replaced “predominantly” with “primarily” (lines 530, 598)

·        We replaced “Looking ahead” with “Prospectively” (line 604)

·        We replaced “facilitating targeted research and more rational clinical decision-making” with supporting focused investigation and evidence-based clinical decision-making (line 620)

·        We replaced “bridge this gap” with “narrow the gap” (lines 94 and 625)

·        We replaced “personalized interventions” with “individualized therapeutic approaches” (line 35) and “personalized therapeutic approaches” (line 627)

Reviewer 2 Comment:
Tables need consistent formatting and standardization.

Response:
Thank you for pointing this out. We have reviewed and revised all tables to ensure formatting consistency in terms of layout, headings, font style, and reference citation style. All tables now follow a standardized format.

Reviewer 2 Comment:
All research has limitations; therefore, please include a limitations section addressing the constraints of this study.

Response:
Thank you for your suggestion. As this is a narrative literature review rather than a systematic review or original research article, a formal limitations section is not typically included. However, we acknowledge the inherent constraints of narrative reviews, such as potential selection bias and lack of reproducibility, and have briefly addressed these aspects in the conclusion section to provide appropriate context for the scope and interpretation of the findings.

Reviewer 2 Comment:
The manuscript would benefit from additional figures, particularly a comprehensive synthesis figure illustrating metabolic pathways and associated diseases.

Response:
Thank you for this thoughtful recommendation. Given the broad of diseases and pathways covered in this review, we believe an additional synthesis figure may risk oversimplifying the data or duplicating content already provided in the tables. The current tables are designed to comprehensively and clearly convey the gene–pathway–disorder relationships.

Reviewer 2 Comment:
Inconsistencies exist in reference formatting; this requires correction.

Response:
We appreciate your attention to detail. We have double-checked the reference list using a reference management software to correct any inadvertent inconsistencies or formatting issues that may have arisen.

Overall, we express our gratitude to Reviewer 2 for the constructive and encouraging feedback, which has been crucial and beneficial in enhancing the quality of our review.

Reviewer 3 Report

Comments and Suggestions for Authors

Final Revision Summary

1. Supplementing References for Tables 1-3. Provide core references for Tables 1-3. 

2. Optimized Conclusion Section

Highlight lactylation’s pathological role alongside heteroplasmy as core challenges, and emphasize the therapeutic potential of targeting the ARRB1/S100A9 pathway.

Comments on the Quality of English Language

The English could be improved to more clearly express the research.

Author Response

Firstly, we would like to express our gratitude to Reviewer 3 for their insightful suggestions and valuable comments, which have significantly enhanced the overall integrity of our manuscript. Below you may find our responses to each point raised, respectively.

1. Supplementing References for Tables 1–3. Provide core references for Tables 1–3.

Response: Thank you for your thoughtful and constructive comment. Core references have been added to the captions of Tables 1–3 to support and validate the data presented.

2. Optimized Conclusion Section: Highlight lactylation’s pathological role alongside heteroplasmy as core challenges and emphasize the therapeutic potential of targeting the ARRB1/S100A9 pathway.

Response: Thank you for this recommendation. The conclusion section has been revised (lines 630–637) to emphasize both lactylation and heteroplasmy as key challenges. We also highlighted the therapeutic promise of the ARRB1/S100A9 pathway in neurodegenerative mitochondrial disorders.

“Lactylation, an emerging post-translational modification linked to altered metabolic states, has been implicated in modulating gene expression and inflammatory responses, thereby contributing to the progression of neurological pathology in mitochondrial diseases [82]. Moreover, therapeutic delivery to affected neural tissues remains a major obstacle. Promisingly, recent findings have highlighted the ARRB1/S100A9 signaling axis as a potential therapeutic target capable of modulating mitochondrial stress responses and neuroinflammation”

Once again, we thank Reviewer 3 for the insightful and constructive feedback, which helped us improve the clarity, coherence, and scientific rigor of our manuscript.

Reviewer 4 Report

Comments and Suggestions for Authors

Major points:

The authors tried to bring genetic, clinical and molecular data to elucidate shared and distinct pathophysiological features. However, this article includes a lot of authors' opinions. I recommend to omit authors opinion.

In abstract and text, heteroplasmy is included. A more detailed description of heteroplasmy, including its problems and cases, will be necessary.

Minor points:

Line 66-81: For instance ----in neuropathology [2]. This is In Introduction part. Please move to 2. Mitochondrial Disorders Categorized by Metabolic Pathways,

Figure needs more explanation. Please add them.

These words will include the author's intention. I recommend using more scientific terms.

the Era of Precision Medicine, treatment strategies, Targeted, a shift from symptom-based care to treatments grounded in molecular etiology, remains limited due to the technical challenge of importing guide RNAs into mitochondria, predominantly, Looking ahead, the potential to develop predictive models, conceptual organization, facilitating targeted research and more rational clinical decision-making, bridge this gap, personalized interventions, both clinical outcomes and quality of life.

Author Response

Firstly, we would like to express our gratitude to Reviewer 4 for their insightful suggestions and valuable comments, which have significantly enhanced the overall integrity of our manuscript. Below you may find our responses to each point raised, respectively.

1. In abstract and text, heteroplasmy is included. A more detailed description of heteroplasmy, including its problems and cases, will be necessary.

Response: Thank you for this insightful suggestion. We expanded the discussion of heteroplasmy in lines 71-74, addressing its biological implications, diagnostic challenges, and relevance to disease heterogeneity.

“Heteroplasmy, the coexistence of normal and mutated mitochondrial DNA within cells, contributes to the unpredictable severity and distribution of symptoms across individuals and tissues. The proportion of mutant mtDNA can vary across tissues and over time, influencing disease severity, age of onset, and symptom heterogeneity”

2. Lines 66–81: For instance ----in neuropathology [2]. This is in the Introduction part. Please move to 2. Mitochondrial Disorders Categorized by Metabolic Pathways.

Response: Thank you for the suggestion. As advised, we have relocated this section to the more appropriate subsection titled “Mitochondrial Disorders Categorized by Metabolic Pathways” to enhance structural clarity and thematic coherence. The revised text now appears in lines 119–137.

3. Figure needs more explanation. Please add them.

Response: Thank you for your constructive comment. We have expanded the figure legend to provide a clearer explanation of the elements and rationale behind the design, facilitating better understanding for readers.

Lines 240-250 Figure 1. Overview of the oxidative phosphorylation (OXPHOS) pathway within the mitochondrial inner membrane, highlighting the five multi-subunit enzyme complexes (I–V) responsible for ATP production. Electrons derived from NADH and FADHâ‚‚ are transferred through complexes I–IV, generating a proton gradient that drives ATP synthesis at complex V (ATP synthase). Molecular oxygen (Oâ‚‚) serves as the terminal electron acceptor, forming water (Hâ‚‚O) at complex IV. Below each complex, nuclear and mitochondrial genes most commonly associated with mitochondrial disorders are listed. Mutations in these genes disrupt specific steps in the OXPHOS pathway, impairing ATP production and leading to energy deficits that particularly affect high-demand tissues such as the brain. Created with BioRender.com.”

4. These words will include the author’s intention. I recommend using more scientific terms: the Era of Precision Medicine, treatment strategies, Targeted, a shift from symptom-based care to treatments grounded in molecular etiology, remains limited due to the technical challenge of importing guide RNAs into mitochondria, predominantly, Looking ahead, the potential to develop predictive models, conceptual organization, facilitating targeted research and more rational clinical decision-making, bridge this gap, personalized interventions, both clinical outcomes and quality of life.

Response: We appreciate the suggestion to elevate the scientific tone. Accordingly, we have revised the phrasing to replace general expressions with more scientifically accurate terminology that better reflects the manuscript’s intent. These changes aim to improve the precision and academic rigor of the text. The following revisions have been made:

  • We replaced “treatment strategies” with “therapeutic approaches”(line 568)
  • We replaced “targeted” with “precision” (line 85), “focused” (line 620)
  • We replaced “a shift from symptom-based care to treatments grounded in molecular etiology” with “a transition from symptomatic management to etiology-informed therapies”(line 571)
  • We replaced “remains limited due to the technical challenge of importing guide RNAs into mitochondria” with “is constrained by current limitations in mitochondrial RNA delivery technologies”(line 576)
  • We replaced “predominantly” with “primarily”(lines 530, 598)
  • We replaced “Looking ahead” with “Prospectively”(line 605)
  • We replaced “potential to develop predictive models”with “opportunity to establish computational prediction” (line 607)
  • We replaced “conceptual organization” with “classification”(line 619)
  • We replaced “facilitating targeted research and more rational clinical decision-making”with supporting focused investigation and evidence-based clinical decision-making (line 620)
  • We replaced “bridge this gap”with “narrow the gap” (lines 94 and 625)
  • We replaced “personalized interventions”with “individualized therapeutic approaches” (line 36) and “personalized therapeutic approaches” (line 627)

Once again, we thank Reviewer 4 for the insightful and constructive feedback, which helped us improve the clarity, coherence, and scientific rigor of our manuscript.

Round 2

Reviewer 3 Report

Comments and Suggestions for Authors

Further Revisions Required: Thorough Proofreading Needed:

1 The manuscript has undergone extensive revisions. The English language should now be carefully proofread.

2 Verification of Table References: The authors have supplemented references for Tables 1-3. Please confirm that the cited references precisely correspond to the specific content presented in each table to ensure clarity and avoid confusion for readers.

Comments on the Quality of English Language

The English could be improved to more clearly express the research.

Author Response

Firstly, we would like to express our gratitude to Reviewer 3 for their insightful suggestions and valuable comments, which have significantly enhanced the overall integrity of our manuscript. Below you may find our responses to each point raised, respectively.

Comment 1: The manuscript has undergone extensive revisions. The English language should now be carefully proofread.

Response 1: We appreciate this feedback. The manuscript has undergone a comprehensive grammatical review by a native English speaker with academic writing expertise. A second round of thorough proofreading was conducted to enhance clarity, consistency, and overall language quality.

Comment 2: Verification of Table References: The authors have supplemented references for Tables 1-3. Please confirm that the cited references precisely correspond to the specific content presented in each table to ensure clarity and avoid confusion for readers.

Response 2: Thank you for your valuable comment. We have carefully reviewed the references cited in relation to Tables 1–3. We confirm that all the references included in the revised manuscript accurately correspond to the specific content presented in each respective table. These references have been selected to ensure clarity and relevance, and we have taken care to align them appropriately to avoid any confusion for readers. We appreciate your attention to detail and the opportunity to improve the clarity of our work.

Once again, we thank Reviewer 3 for the insightful and constructive feedback, which helped us improve the clarity, coherence, and scientific rigor of our manuscript.